# Carbonate production of Micronesian reefs suppressed by thermal anomalies and *Acanthaster* as sea-level rises

**Robert van Woesik** *, **Christopher William Cacciapaglia**

Institute for Global Ecology, Department of Ocean Engineering and Sciences, Florida Institute of Technology, Melbourne, Florida, United States of America

* rvw@fit.edu

**Data Availability Statement:** The authors confirm that all data underlying the findings are fully available without restriction. All data and R-code

## Abstract

Coral reefs are essential to millions of island inhabitants. Yet, coral reefs are threatened by thermal anomalies associated with climate change and by local disturbances that include land-use change, pollution, and the coral-eating sea star *Acanthaster solaris*. In combination, these disturbances cause coral mortality that reduce the capacity of reefs to produce enough carbonate to keep up with sea-level rise. This study compared the reef-building capacity of shallow-water inner, patch, and outer reefs in the two islands of Pohnpei and Kosrae, Federated States of Micronesia. We identified which reefs were likely to keep up with sea-level rise under different climate-change scenarios, and estimated whether there were differences across habitats in the threshold of percentage coral cover at which net carbonate production becomes negative. We also quantified the influence of *A. solaris* on carbonate production. Whereas the northwestern outer reefs of Pohnpei and Kosrae had the highest net rates of carbonate production (18.5 and 16.4 kg CaCO$_3$ m$^{-2}$ yr$^{-1}$, respectively), the southeastern outer reefs had the lowest rates of carbonate production (1.2–1.3 and 0.7 kg CaCO$_3$ m$^{-2}$ yr$^{-1}$, respectively). The patch reefs of Pohnpei had on average higher net carbonate production rates (9.5 kg CaCO$_3$ m$^{-2}$ yr$^{-1}$) than the inner reefs of both Pohnpei and Kosrae (7.0 and 7.8 kg CaCO$_3$ m$^{-2}$ yr$^{-1}$, respectively). *A. solaris* were common on Kosrae and caused an average reduction in carbonate production of 0.6 kg CaCO$_3$ m$^{-2}$ yr$^{-1}$ on Kosraean reefs. Northern outer reefs are the most likely habitats to keep up with sea-level rise in both Pohnpei and Kosrae. Overall, the inner reefs of Pohnpei and Kosrae need ~ 5.5% more coral cover to generate the same amount of carbonate as outer reefs. Therefore, inner reefs need special protection from land-use change and local pollution to keep pace with sea-level rise under all climate-change scenarios.

## Introduction

Coral reefs are an integral component of global marine ecosystems and are essential to millions of people that benefit from the goods and services that coral reefs provide. For example, coral reefs reduce storm-wave energy by up to 97% [1], reducing the threat of coastal inundation

files are available in the Supporting Information files.

**Funding:** RvW received funding for the research from the National Science Foundation, award NSF OCE-1657633. The funders did not play a role in the study design, data collection, data analysis, decision to publish, and preparation of the manuscript.

**Competing interests:** The authors have declared that no competing interests exist.

during severe storms [2]. However, contemporary thermal-stress events, associated with global-climate change, cause coral bleaching and mortality, which can lead to shifts in species dominance [3–6]. These changes have reduced the capacity of coral reefs to accrete carbonate in some localities and keep up with sea-level rise [7]. Here we examine whether the reefs of Pohnpei and Kosrae, Federated States of Micronesia (FSM; Figure A in S1 File), are producing enough carbonate to keep up with sea-level rise, while experiencing thermal stress and local disturbances.

Rates of carbonate production have been studied using geological coring [8–10], hydro-chemistry [11], modeling [12], and *in situ* estimates [13–15]. All approaches show considerable variation across ocean basins, with erosional forces only exceeding rates of carbonate production when gross calcification rates are low. In a global synthesis, Vecsei (2004) [16] showed that carbonate production decreased with depth and was lower on reef flats than in other habitats. These results agree with van Woesik and Cacciapaglia (2018) [17] who showed major differences in carbonate production among reef habitats in the Republic of Palau and the island of Yap, western FSM, with outer reefs averaging greater carbonate production (i.e., 10 kg $CaCO_3$ $m^{-2}$ $yr^{-1}$) than inner reefs (i.e., averaging 7 kg $CaCO_3$ $m^{-2}$ $yr^{-1}$). Yet, in less favorable environments, Perry et al. (2013) [7] estimated that Caribbean reefs have modern net carbonate production rates averaging only 1.5 kg $CaCO_3$ $m^{-2}$ $yr^{-1}$, with some reefs showing net negative carbonate budgets. On the shallow fore-reef slopes in the Maldives in the Indian Ocean, Perry and Morgan (2017) [18] showed that after a thermal-stress event net accretion rates were negative, at -3 kg $CaCO_3$ $m^{-2}$ $yr^{-1}$. Although, through the same thermal-stress event, Ryan et al. (2019) [15] showed evidence that the upper reef crest and reef flats of the same Maldivian reef maintained positive accretion rates. In a study in the Seychelles, Januchowski-Hartley et al. (2017) [19] showed that carbonate production was dependent on thermal stress, depth, macro-algal presence, wave energy, and the abundance of excavating parrotfish.

Carbonate production rates can be influenced also by other chronic local disturbances. For example, the coral predator *Acanthaster solaris* has long been known to reduce coral populations when the sea stars are in high densities [20]. While high densities of *Acanthaster* larvae have been associated with elevated nutrient concentrations through river discharge [21, 22], other aspects of their biology and ecology remain unresolved [23]. Here we examine the influence of *A. solaris* populations on the capacity of Pohnpeian and Kosraean shallow-water reefs to produce carbonate, and determine the density of *A. solaris*, relative to the available percentage of live coral cover, beyond which carbonate production is reduced to zero.

Quantifying carbonate production is critical when predicting how coral reef systems will respond to sea-level rise, especially as the rate of sea-level rise is predicted to accelerate rapidly into the future [24–26]. These estimates in carbonate production should also influence conservation targets, especially if inner reefs require more coral cover to produce the same amount of carbonate as outer reefs [17]. Here we examine the coral reefs on two islands, Pohnpei and Kosrae, FSM, and quantify the *in situ* rates of carbonate production to identify which reefs are likely to keep up with sea-level rise [27, 28] under different climate-change scenarios. We also estimate whether there are differences in the threshold of percentage coral cover, at which net carbonate production becomes negative, across habitats.

## Methods

### Study design and field methods

Twenty-four study sites were randomly selected in each of Pohnpei (6.2˚N, 158.2˚E) and Kosrae (5.3˚N, 162.9˚E) FSM using a randomly stratified sampling approach with the package *sp* [29] in R [30]. In Pohnpei, reefs were stratified as inner reefs, patch reefs, and outer reefs. In

Kosrae, we only stratified the reefs as either inner reefs or outer reefs (because of the lack of patch reefs). Sample size of each strata was determined by calculating the geographic area of each reef type, using the *area* function from the R package *raster* [31], and allocating the number of sites in accordance with the area estimates. Reef surveys focused on the 2–5 meters depth contour to estimate shallow-water carbonate production.

Six, 10 m transects, using a modified line-intercept technique that followed the reef substrate, were used to measure the benthic composition for every centimeter, at each site of the 48 sites [32, 17]. A few meters gap was allocated between the ends of the transects to ensure no overlap of substrate between transects. Corals were recorded to species level, except massive *Porites* and encrusting *Montipora*, which were recorded in the field as growth forms. All other organisms along each transect were identified to the highest possible taxonomic resolution. Rugosity was recorded using the planar length of a second transect that spanned across the reef horizontally. Echinoids were recorded within 30 cm on either side of the 10 m tape. The urchins were recorded as *Echinometra*, *Diadema*, and 'Other', and the diameter of each echinoid test was measured to the nearest 0.5 cm. The abundance of *Acanthaster solaris* (crown-of-thorns sea star) were recorded within 5 m along each of the six 10 m transects. Herbivorous parrotfishes were videoed and identified to species and their estimated length was recorded to the nearest cm along six transects, each of which was 30 m long by 4 m wide. Care was taken to record the fish-transect videos ahead of the other transects to avoid any disturbance to the fishes.

## Carbonate production

Net carbonate production (kg CaCO$_3$ m$^{-2}$ yr$^{-1}$) was estimated using the following equation:

$$Carbonate\ production_i = calcification_i + sgn(x)sedimentation_i - erosion_i - Acanthaster_i \quad (1)$$

where *calcification* is the gross carbonate production by reef building organisms at site *i* [33]; *sedimentation* is the contribution of sediment to the reef, where it increases carbonate production rates if sedimentation is low ($< 0.05$ kg m$^{-2}$ d$^{-1}$) [33,34] and then *sgn (x)* is positive, whereas if terrestrial sedimentation is high *sgn (x)* is negative because the sediment smothers corals; *erosion* is the rate of erosion, estimated following van Woesik (2013) [33]; and *Acanthaster* is amount of carbonate potentially lost by *Acanthaster solaris* (i.e., the crown-of-thorns sea star) eating corals. High densities of *A. solaris* reduce live coral cover [35], which in turn reduces a reef's capacity to produce carbonate. *Calcification* of organisms was calculated as follows:

$$calcification_i = r_i \left\{ \Sigma \left[ \left( m_{i,j} * \frac{x_{i,j}}{100} \right) * d_{i,j} * g_{i,j} * 10 \right] + ca_i \right\} \quad (2)$$

where *r* is the rugosity of site *i* averaged across six transects; *m* is the adjustment coefficient for the morphology of species *j* at site *i* (following van Woesik and Cacciapaglia 2018) [17]; *x* is the planar percent cover averaged across site *i* for species *j*, *d* is the density (g cm$^{-3}$) of species *j* (following [17]) in site *i*; *g* is the vertical growth rate of coral species j (cm yr$^{-1}$) (after van Woesik and Cacciapaglia 2018 [17]); 10 is an adjustment constant to convert units back to kg CaCO$_3$ m$^{-2}$ yr$^{-1}$; and *ca* is the contribution of coralline algae to carbonate production, calculated following Perry et al. (2012) [14] as:

$$ca_i = 0.018 * (pca_i) * 10 \quad (3)$$

where *pca* is the planar coralline algae cover averaged across six transects at site *i*, 0.018 is the gross carbonate production (g cm$^{-2}$ yr$^{-1}$) estimated using averages from Perry et al. (2012) [14]; and 10 is an adjustment constant used to convert units from g cm$^{-2}$ yr$^{-1}$ to kg m$^{-2}$ yr$^{-1}$.

## Sedimentation

The accretion of reefs can be supplemented by calcareous sedimentation [9,10], or compromised by excessive amounts of terrestrial sedimentation (when $> 0.05$ kg m$^{-2}$ d$^{-1}$) [34], which causes coral smothering and reduces the rate of carbonate production [36]. The sedimentation rate that was used in Pohnpei and Kosrae was 0.4 kg CaCO$_3$ m$^{-2}$ yr$^{-1}$ following estimates from Montaggioni (2005) [10] and Hubbard (1997) [36]. We witnessed some terrestrial runoff and a high deposition of fine sediment in the southern bay of Kosrae (Utwe Bay), and we therefore introduced a negative sedimentation component to Eq 1, using -0.4 kg CaCO$_3$ m$^{-2}$ yr$^{-1}$ [37], at sites that were downstream of river runoff at that location.

## Erosion

Reef erosion was comprised of three biological components, echinoids or sea urchins, herbivorous fishes, and macroboring organisms. Gross erosional rates were calculated as:

$$Erosion_i = \Sigma(parrotfish_{i,j} + urchin_{i,j}) + macroboring_i \qquad (4)$$

where *parrotfish* is the rate of erosion by herbivorous fish species *j* at site *i*; *urchin* is the rate of biological erosion by sea urchins species *j* at site *i*; and *macroboring* is the erosional forces of macroboring organisms in site *i*. *Erosion* by parrotfish was estimated after [14] using the equation:

$$parrotfish_i = \Sigma(vol_{i,j,n} * sp_{i,j,n} * br_{i,j,n}) * D_i * 365 * 0.001 \qquad (5)$$

where *vol* is the estimated volume of the bites of individual parrotfish *n* for species *j* at site *i*; *sp* is the scar proportion, or the proportion of bites that leave scars on corals for individual *n*, of species *j* at site *i*; *br* is the bite rate (bites day$^{-1}$) of individual *n*, of species *j*, at site *i*; the average density *D* of corals was calculated at site *i* based on coral composition; the constant *365* was to convert days into years; and *0.001* was a constant to convert grams into kilograms. Bite volume *vol* was further defined using the following equation:

$$vol_{i,j,n} = \frac{e^{1.32+0.06*length_{i,j,n}}}{1000} \qquad (6)$$

where *length* is the length (cm) of each parrotfish n, of species j at site *i*; the constants were gained using a linear regression of data collected by Ong and Holland (2010) [38]; the constant one thousand was used to convert cubic millimeters to cubic centimeters. Scar proportion, *sp*, from Eq 5 was further defined as follows:

$$sp_{i,j,n} = 1/[1 + e^{-(-2.46+0.089*length_{i,j,n})}] \qquad (7)$$

where *length* is the length of fish *n* of species *j* at site *i*. The equation was based on a regression using data from Bonaldo and Bellwood (2008) [39] and Ong and Holland (2010) [38]. Bite rate, *br*, from Eq 5 was defined as:

$$br_{i,j,n} = 60\left\{[(4.31 + brc_{i,j} - 0.36) - (0.045 * reeftime * length_{i,j,n})]\right\} \qquad (8)$$

where *brc* is a bite rate constant, *reeftime* is the amount of time fishes spend grazing on reefs, estimated to be 9 hours per day. These constants were estimated by Peter Mumby (personal communication). *Length* is length of fish *n* of species *j* at site *i*.

Erosion by sea urchins (kg $CaCO_3$ $m^{-2}$) was estimated after [14] using the following equation:

$$urchin_i = \Sigma(Diadema_{i,n} + Echinometra_{i,n} + Other\ urchins_{i,n}) \tag{9}$$

where *Diadema* is the erosion caused by a *Diadema* individual *n* at site *i*; *Echinometra* is the erosion caused by an *Echinometra* individual *n* at site *i*, and *Other urchins* is the erosion caused by sea urchins that were not *Echinometra* or *Diadema*. *Diadema* was defined as:

$$Diadema_{i,n} = (0.000001 * diameter_{i,n}^{3.42}) * 0.365 * 0.57 \tag{10}$$

following an equation by Januchowski-Hartley et al. (2017) [19], where *diameter* is the test size (cm) of the individual *n* at site *i*. *Echinometra*, from Eq 9, also follows an equation from Januchowski-Hartley et al. (2017) [19]:

$$Echinometra_{i,n} = (0.0004 * diameter_{i,n}^{1.98}) * 0.365 * 0.57 \tag{11}$$

where *diameter* is the test size of individual *n* within the genus *Echinometra* at site *i*. *Other urchins* in Eq 9 follows an equation from Januchowski-Hartley et al. (2017) [19], as follows:

$$Other\ urchins_{i,n} = (0.0001 * diameter_{i,n}^{2.32}) * 0.365 * 0.57 \tag{12}$$

where *diameter* is the test size of individual *n* outside the genus of *Echinometra* or *Diadema* at site *i*. Macroboring organisms were included into Eq 4 to incorporate the erosional forces of boring sponges following the equation:

$$macroboring_i = plamc_i * mec \tag{13}$$

where *plamc* is the planar cover of the macroboring organisms averaged over at site *i*; and *mec* is the constant used to define macroboring erosion, which was set as a conservative 10 kg $CaCO_3$ $m^{-2}$ yr-1 following Glynn (1997) [40].

## Acanthaster

While Glynn (1973) [41] estimated the densities of *Acanthaster* that would overwhelm the ability for corals to persist, there have been no studies aimed at quantifying the influence of *Acanthaster* on carbonate production rates. We used field estimates to evaluate the effect of *Acanthaster solaris* on carbonate production as follows:

$$Acanthaster_i = (tc.transect_i * RI_i)/50 \tag{14}$$

where $Acanthaster_i$ is the reduction in gross production caused by observed *A. solaris* at site *i*; $RI_i$ is the rate of coral ingestion at site *i* (see Eq 16); *50* is a constant to convert observational transect size to $m^2$; and $tc.transect_i$ is the per transect consumption rate:

$$tc.transect_i = A.sp_i * con * Density_i * 10 * 365 \tag{15}$$

where $A.sp_i$ is the number of *A. solaris* observed in site *i* divided by the number of transects; *con* is the average consumption rate (0.01 $m^2$ $d^{-1}$) estimated from Keesing and Lucas (1992) [42]; $Density_i$ is the average density of corals in site *i* (g $cm^3$); *10* is the constant used to convert the unit to kg $m^2$, and *365* converts days to years.

$$RI_i = \frac{a_i * R_i}{1 + a_i * h * R_i} * 4.53 \tag{16}$$

where $RI_i$ is the rate of coral ingestion in site *i*; $R_i$ is the resource density or live coral cover (%)

at site *i*. The handling rate, or how long it takes for a single *A. solaris* to eat a coral colony, *h*, was estimated using the average size of coral colonies and the average rate of consumption, which was conservatively estimated to be around 3.5 days. The *4.53* constant is used to rescale $RI_i$, which is resource dependent, to match average coral density with estimated consumption rate; $a_i$ is the attack rate, estimated using the speed at which *A. solaris* can move and the density of corals in the transect following Eq 17:

$$a_i = 12/\{([([100 - R_i + 1_E10]/10)/2])^2 * pi\}/speed \tag{17}$$

where *12* is a constant for active predatory hours; $R_i$ is the resource density or live coral cover in site *i* as a percentage, subtracted from 100; and $1_E10$ was added to convert the value to a non-zero area where corals are present. These values are divided by *10* for the transect length, to determine average distance between corals, and it was assumed the *A. solaris* had to search the area of a circle with this average distance between corals equaling the diameter of that circle. The area was then divided by the speed at which *A. solaris* can move, *speed*, (504 m d$^{-1}$; Muller et al. 2011 [43]). It was assumed that *A. solaris* could only reduce or negate carbonate production in this model, so the effects were subtracted from gross production to a maximum erosional force netting zero gross production.

To convert net carbonate production, from Eq 1, to vertical reef growth (in mm) we used:

$$Vertical\ reef\ accretion = Cp + Cp(Cp * alpha) \tag{18}$$

where *Cp* is carbonate production (from Eq 1) and *alpha* is a coefficient estimated as -0.01949 (after van Woesik and Cacciapaglia 2018 [17]).

## Carbonate thresholds

We used an additive mixed effects model in a Bayesian framework [44] to estimate the value of coral cover, for the different habitats, at which net carbonate production became negative, using the following:

$$G = Beta + f(coral\ cover) + Habitat + a + error \tag{19}$$

where *G* is the net carbonate production at site *i*; *f(coral cover)* uses an O'Sullivan spline smoothing function [45]; *Habitat* is the fixed effect of interest; *a* is a random intercept for site; and *error* is the error term for the residuals. We assumed that no prior information was known and therefore used multivariate normal diffuse and normal diffuse priors [44]. All models were run in R and coded in JAGS [46] (all the R code is available in S1 Data and at https://github.com/rvanwoesik).

We would also like to thank Eugene Joseph the Director of the Conservation Society of Pohnpei and Andy George the Director of the Kosrae Conservation Society for granting us permission to conduct research on Pohnpei and Kosrae respectively.

## Results

Gross carbonate production on Pohnpei averaged 8.2 kg CaCO$_3$ yr$^{-1}$, and was on average higher on patch reefs (9.1 kg CaCO$_3$ m$^{-2}$ yr$^{-1}$) than on outer reefs (7.7 kg CaCO$_3$ m$^{-2}$ yr$^{-1}$) and on inner reefs (6.8 kg CaCO$_3$ m$^{-2}$ yr$^{-1}$) (Table 1). Net carbonate production rates closely followed rates of gross production (Table 1), although within-habitat differences were considerable (Figs 1 and 2). For example, the outer northwestern reefs of Pohnpei supported the highest rates of net carbonate production (18.5 kg CaCO$_3$ m$^{-2}$ yr$^{-1}$), and the lowest rates were recorded on the southeastern outer reefs (1.2–1.3 kg CaCO$_3$ m$^{-2}$ yr$^{-1}$), (Table 1 and Fig 2).

**Table 1. Carbonate production and erosion rates across shallow-water coral reef habitats (2–5 m depth) on Pohnpei and Kosrae, Federated States of Micronesia.**
All values are in kg CaCO$_3$ m$^{-2}$ yr$^{-1}$. Gross production is the total rate of carbonate production across habitats, excluding erosion rates. Erosion includes parrotfish and urchin erosional forces combined (which does not include *Acanthaster solaris* erosion). Net production is gross carbonate production minus the erosional estimates and sedimentation inputs. *Acanthaster* erosion is the erosion caused by *A. solaris*. Combined reef strata were only available for inner and outer reefs, due to the lack of patch reefs in Kosrae.

| Coral-reef habitat | Gross production (kg CaCO$_3$ m$^{-2}$ yr$^{-1}$) | Erosion rates (kg CaCO$_3$ m$^{-2}$ yr$^{-1}$) | Net production (kg CaCO$_3$ m$^{-2}$yr$^{-1}$) | Acanthaster erosion (kg CaCO$_3$ m$^{-2}$yr$^{-1}$) |
|---|---|---|---|---|
| *Pohnpei (combined)* | 8.17 ± 1.77 | 0.04 ± 0.04 | 8.49 ± 1.75 | 0.04 ± 0.05 |
| *Pohnpei inner* | 6.75 ± 3.60 | 0.01 ± 0.01 | 7.10 ± 3.60 | 0.03 ± 0.05 |
| *Pohnpei patch* | 9.12 ± 2.04 | 0.01 ± <0.01 | 9.47 ± 2.04 | 0.00 |
| *Pohnpei outer* | 7.74 ± 4.07 | 0.09 ± 0.11 | 8.01 ± 4.0 | 0.09 ± 0.14 |
| *Kosrae (combined)* | 7.42 ±1.40 | 0.04 ± 0.03 | 7.51 ± 1.41 | 0.61 ± 0.50 |
| *Kosrae inner* | 6.95 ±2.14 | 0.01 ± <0.01 | 6.98 ± 2.0 | 0.28 ± 0.36 |
| *Kosrae outer* | 7.56 ± 1.76 | 0.04 ± 0.03 | 7.69 ± 1.77 | 0.72 ± 0.65 |
| *Both* islands inner | 6.86 ±1.90 | 0.01 ± <0.01 | 7.03 ± 1.86 | 0.15± 0.19 |
| *Both* islands outer | 7.62±1.70 | 0.06 ± 0.04 | 7.79 ± 1.69 | 0.52 ± 0.46 |

Gross carbonate production on Kosrae averaged 7.4 kg CaCO$_3$ m$^{-2}$ yr$^{-1}$, although on average the outer and inner reefs did not vary greatly (7.6 and 7.0 kg CaCO$_3$ m$^{-2}$ yr$^{-1}$, respectively). Similar to Pohnpei, carbonate production on the outer reefs on Kosrae was variable and was highest on the northern outer reefs (16.4 kg CaCO$_3$ m$^{-2}$ yr$^{-1}$), and lowest on southeastern outer reefs (0.7 kg CaCO$_3$ m$^{-2}$ yr$^{-1}$) (Table 1 and Fig 3). Net production differed from gross

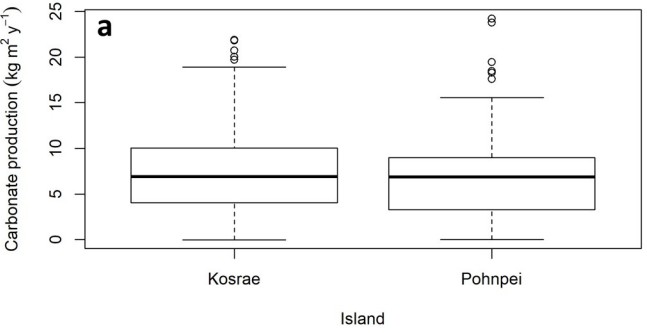

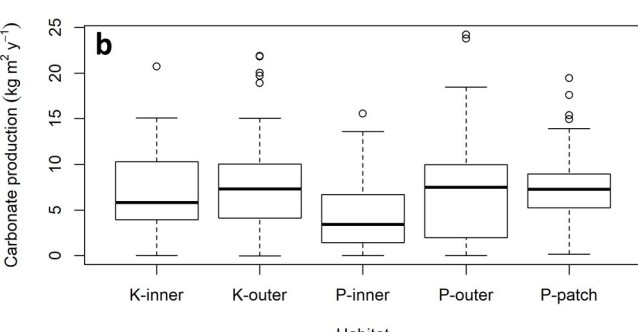

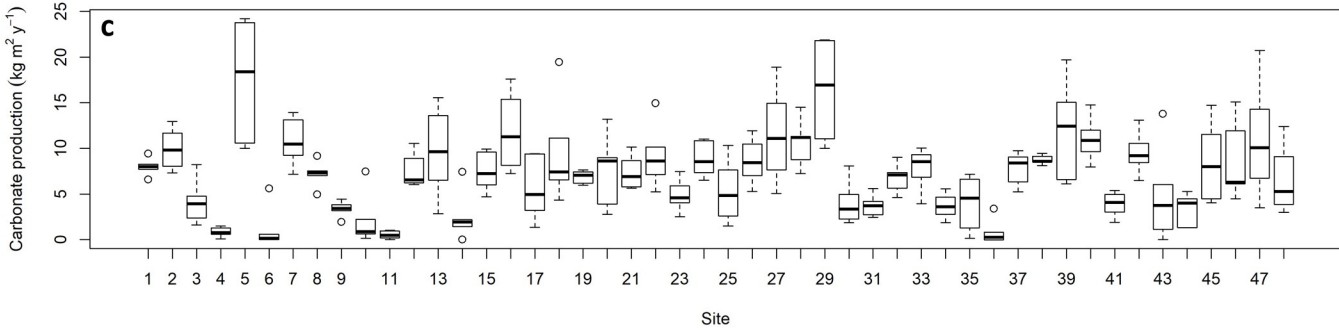

**Fig 1.** Net shallow-water coral-reef carbonate production stratified by (a) island, (b) habitat type (at both Kosrae and Pohnpei inner reefs and outer reefs, and Pohnpei patch reefs), and (c) site (at 24 sites in Pohnpei and 24 sites in Kosrae, Federated States of Micronesia) 2018. The thick horizontal lines are the medians, the box surrounding the medians are the first and third quartiles, the whiskers identify the range of the data, and the circles identify outliers. These data do not include the erosional effects of *Acanthaster solaris*.

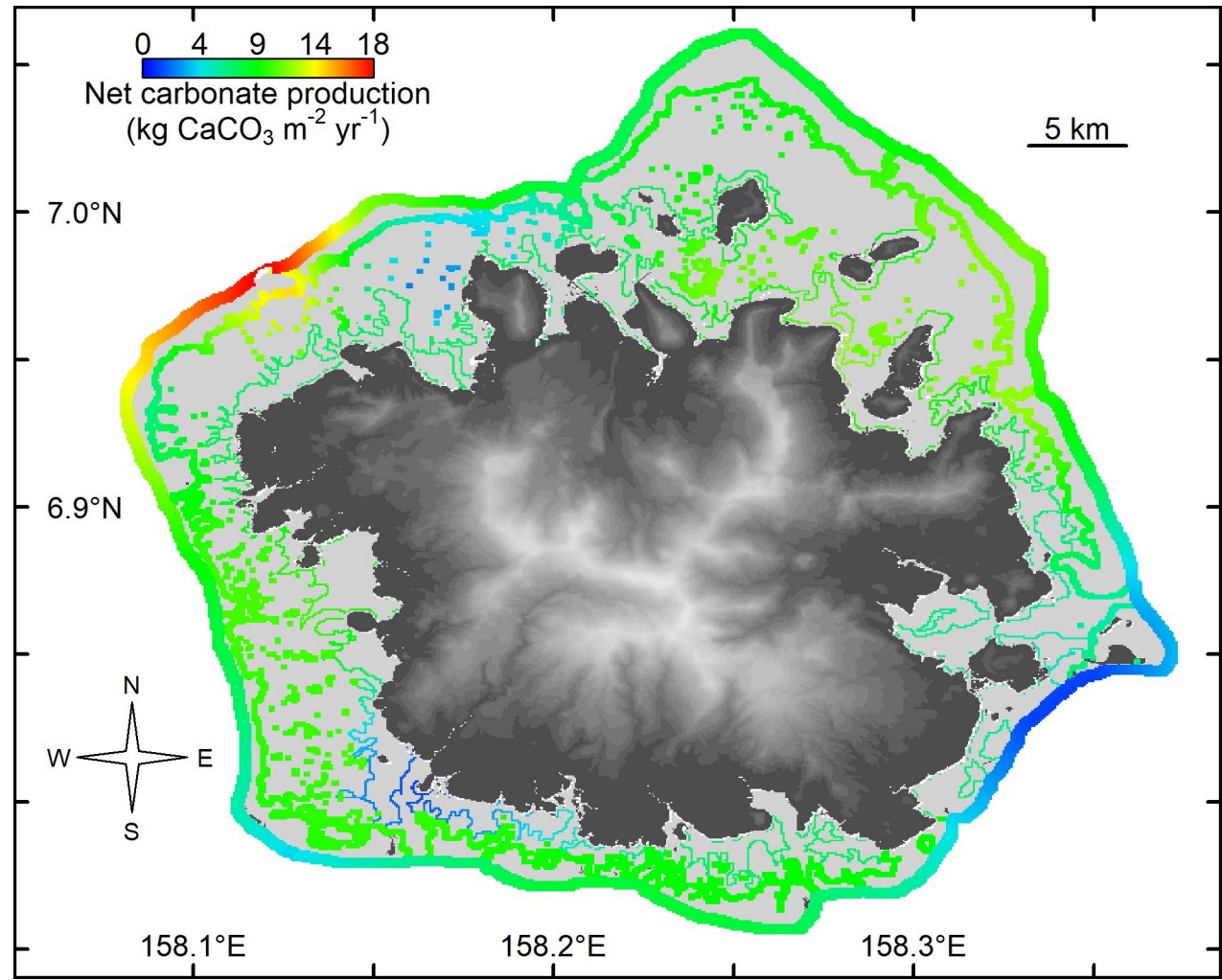

**Fig 2. Spatial kriging of the net shallow-water coral-reef carbonate production (kg CaCO$_3$ m$^{-2}$ yr$^{-1}$) without the influence of *Acanthaster*, for 24 sites in Pohnpei, Federated States of Micronesia, 2018.** Base elevation map was plotted in R using raw 10 m Digital Elevation Model from https://pae-paha.pacioos.hawaii.edu/thredds/ncss/usgs_dem_10m_pohnpei/dataset.html.

production at sites where sedimentation and erosion were much higher than background rates. This occurred in Utwe Bay in southern Kosrae, where terrestrially derived sediment was much higher than elsewhere (personal observations). Terrestrially derived sediment smothers coral colonies and thereby reduces carbonate production.

The reefs of Pohnpei and Kosrae supported similar coral assemblages, although there were some differences in species dominance. The reefs of Pohnpei, particularly the patch and inner reefs, were dominated by *Porites rus*, *Porites cylindrica*, and *Porites lobata*. The outer reefs were dominated by encrusting *Montipora* and *Acropora hyacinthus* (Fig 4). These five species contributed 82% of the gross carbonate production on Pohnpei. The reefs of Kosrae were dominated by encrusting *Montipora*, *Porites rus*, *Goniastrea retiformis*, *Porites lobata*, and *Porites lichen* (Fig 4). These five species contributed 78% of the total gross carbonate production on Kosrae. Importantly, the inner reefs of both Pohnpei and Kosrae, and the patch reefs of Pohnpei had a higher live-coral-cover threshold than the outer reefs of both islands, although

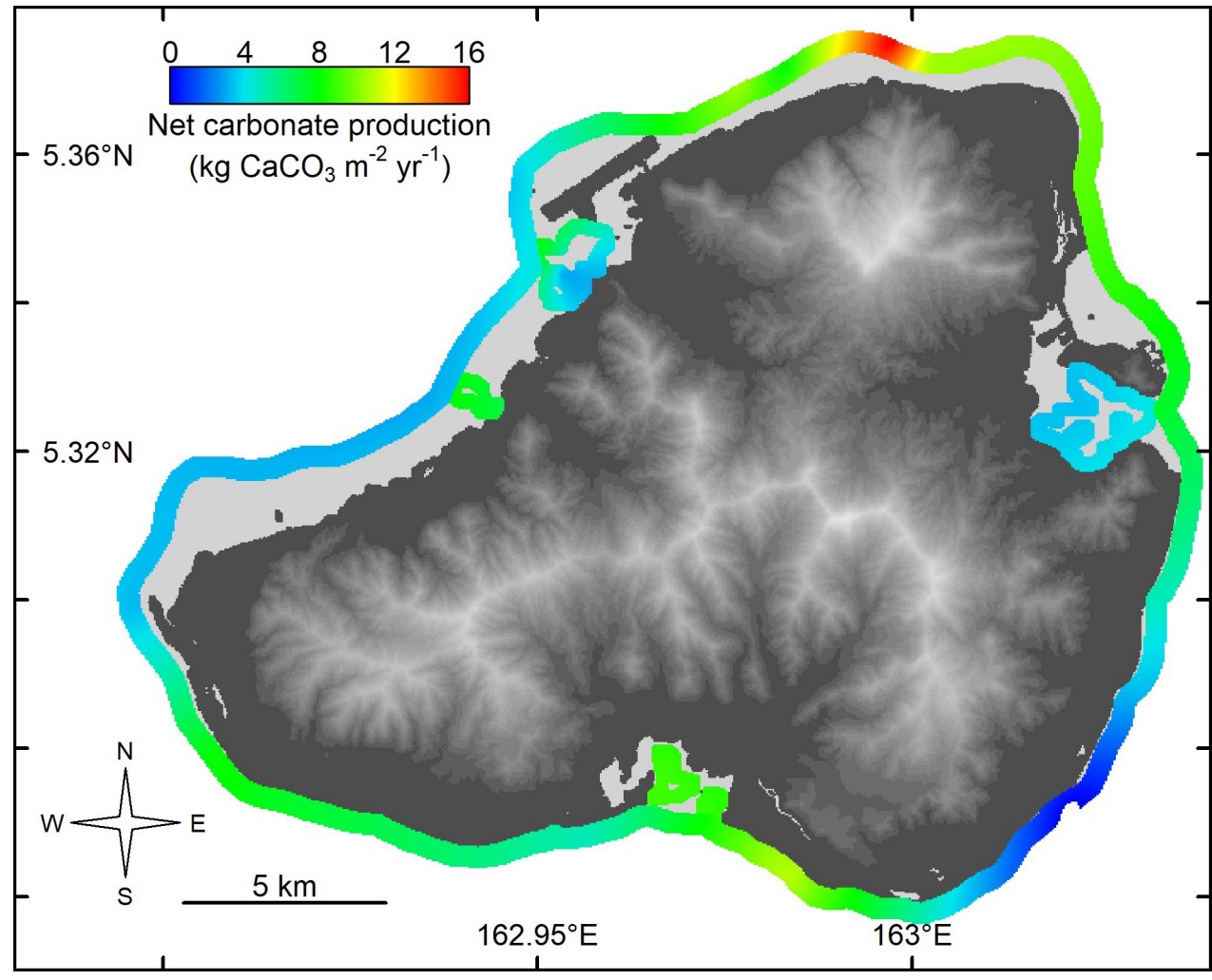

**Fig 3. Spatial kriging of the net shallow-water coral-reef carbonate production (kg CaCO₃ m⁻² yr⁻¹) without the influence of *Acanthaster*, for 24 sites in Kosrae, Federated States of Micronesia, 2018.** Base elevation map was plotted in R using raw 10 m Digital Elevation Model from https://pae-paha.pacioos.hawaii.edu/thredds/ncss/usgs_dem_10m_kosrae/dataset.html.

there was considerable uncertainty in the thresholds for the inner reefs (i.e., high 95% credible intervals) (Fig 5).

*Acanthaster solaris* were observed on reefs of both islands although in 2018 populations indicative of an outbreak (>30 hectare⁻¹) were only observed on some of the shallow reefs of Kosrae. Since outer reefs tended to have the highest densities of *A. solaris*, carbonate production on these outer reefs were most affected (Table 1 and Figs 6 and 7). We re-ran the carbonate production model for both islands to incorporate *A. solaris* and found that carbonate production was reduced on Kosrae by an average 0.6 kg CaCO₃ m⁻² yr⁻¹ and on Pohnpei by 0.04 kg CaCO₃ m⁻² yr⁻¹, across all habitats (Table 1 and Figs 6 and 7). *A. solaris* densities did not reduce carbonate production to negative values, although at two northwestern sites on Kosrae gross carbonate production was reduced by 80% and 62%, where *A. solaris* densities were 17 and 14 (per 300 m²) and where coral cover was low. For mitigation purposes, and to sustain a productive reef, we found that *A. solaris* densities should be kept below a density threshold that is proportional to 7.3% of the relative coral densities (Figure B in S1 File). For

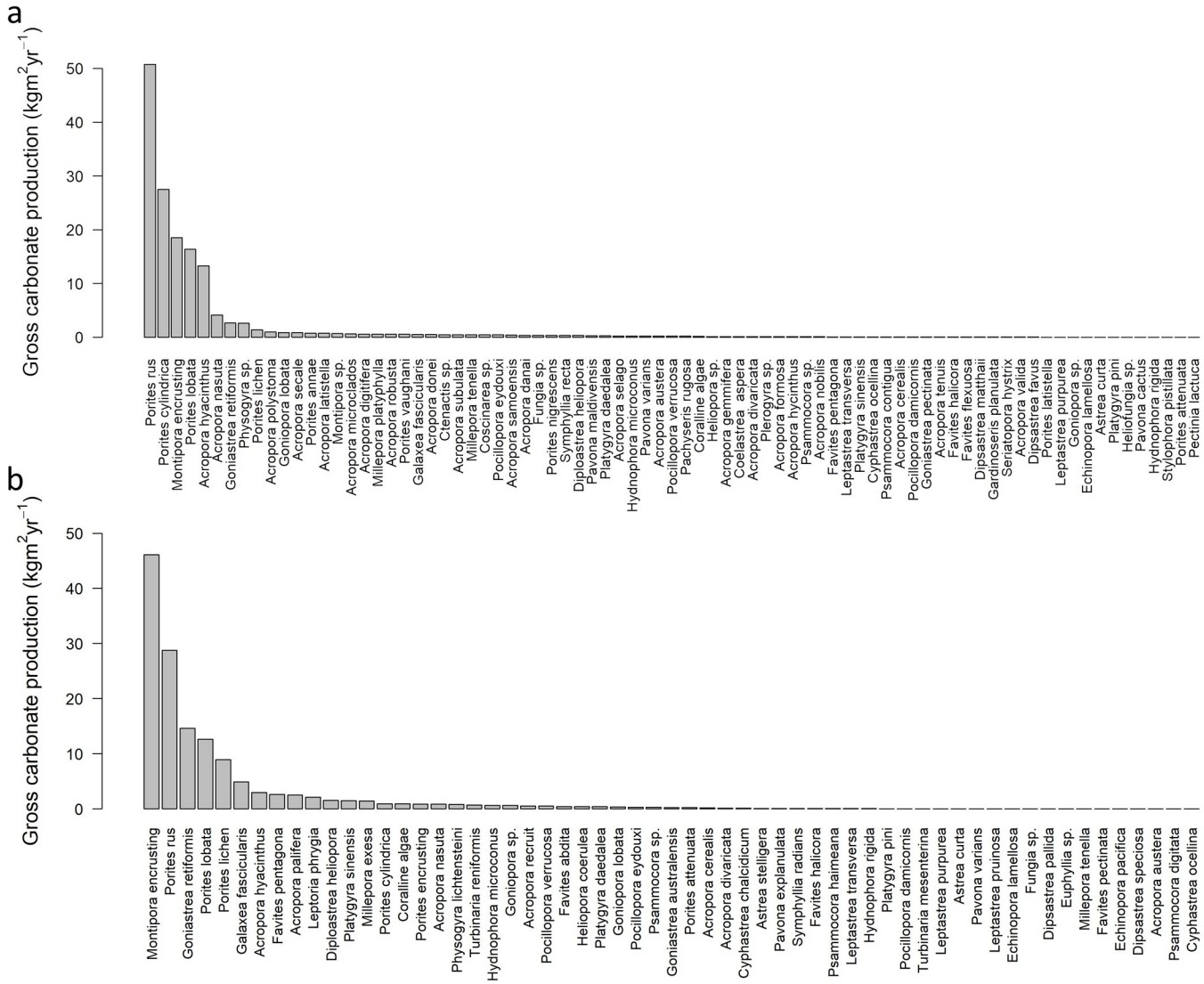

**Fig 4.** Cumulative shallow-water coral-reef carbonate production by coral species and other benthic taxa for (a) 24 sites in Pohnpei and (b) 24 sites in Kosrae, Federated States of Micronesia, 2018.

example, if a 100 m$^2$ site supports 30% live coral cover, any more than two *Acanthaster* in that site for one year will likely reduce gross carbonate production to zero.

Other echinoids also reduce carbonate production, particularly *Echinometra mathaii*, and related species in high densities [38]. We noticed that large populations of *E. mathaii* were common on southeastern outer reefs and caused considerable erosion (Figures C and D in S1 File) and are less common on inner and patch reefs. Carbonate erosion by parrotfishes was also high on the southeastern reefs and along western reefs, and was lower elsewhere (Figures E and F in S1 File).

## Discussion

This study aimed to identify the spatial variation of shallow-water carbonate production in Pohnpei and Kosrae, Federated States of Micronesia, to assess which reefs are likely to keep up with sea-level rise, and to determine what role *Acanthaster solaris* plays in carbonate

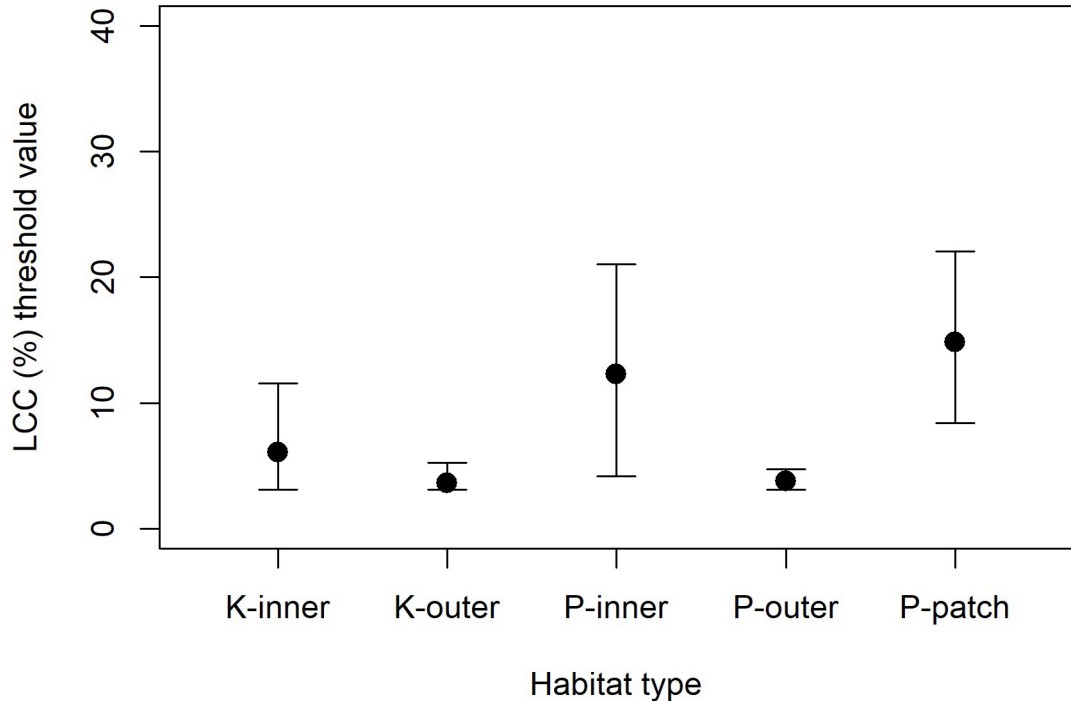

**Fig 5.** Percent threshold live coral cover (LCC) needed to maintain net positive accretion stratified by shallow-water coral-reef habitat (at both Kosrae and Pohnpei inner reefs and outer reefs, and Pohnpei patch reefs) for (a) 24 sites in Pohnpei and (b) 24 sites in Kosrae, Federated States of Micronesia, 2018. The dots represent the posterior means and the vertical lines represent the 95% credible intervals.

production. While the leeward, northern and northwestern facing reefs had the highest rates of net carbonate production (16.5–18.5 kg $CaCO_3$ $m^{-2}$ $yr^{-1}$), the windward, southeastern facing reefs showed the lowest rates of net carbonate production (0.7–1.3 kg $CaCO_3$ $m^{-2}$ $yr^{-1}$). Such high variation in carbonate production along the outer shallow-water reefs is important, especially since habitats with low rates may not have the capacity to keep up with predicted sea-level rise. Based on different greenhouse-gas-emission scenarios, most frequently conveyed as Representative Carbon Pathways (RCPs) 2.6. 4.5, 6.0, and 8.5 $Wm^{-2}$, the predicted rates of sea-level rise by the year 2100 have been conservatively estimated at 5, 6.5, 6.7, and 9 mm $yr^{-1}$, respectively [25]. Converting the field estimated rates of carbonate production to vertical rates of reef accretion (following Eq 18) the northwestern shallow-water outer reefs of Pohnpei and the northern shallow-water outer reefs of Kosrae are estimated to vertically accrete at 11.8 mm $yr^{-1}$ and 11.2 mm $yr^{-1}$, respectively. These rates of vertical accretion are relatively high for contemporary reefs and exceed the rates of sea-level rise under RCP 8.5. Therefore, if effectively managed, the northern shallow-water outer reefs of Pohnpei and Kosrae will likely have the capacity to keep up with sea-level rise and maintain their essential ecosystem functions.

By contrast, the southeastern shallow-water reefs of Pohnpei and Kosrae, have estimated vertical accretion rates of only 1.3 mm $yr^{-1}$. Although we did not directly measure sedimentation rates nor did we measure micro-bioerosion rates, our data suggest that the southeastern reefs are not likely to keep up with sea-level rise by the year 2100. Indeed, the projections of our model show that without considering spatial variation, on average, Pohnpei's and Kosrae's shallow-water outer reefs fall short of the moderate rates of sea-level rise projected under RCP 4.5 (i.e., 6.2 mm $yr^{-1}$ accretion and 6.5 mm $yr^{-1}$ of sea-level rise under RCP 4.5, Figure G in S1 File). Most concerning is that the shallow-water inner reefs of both islands have estimated

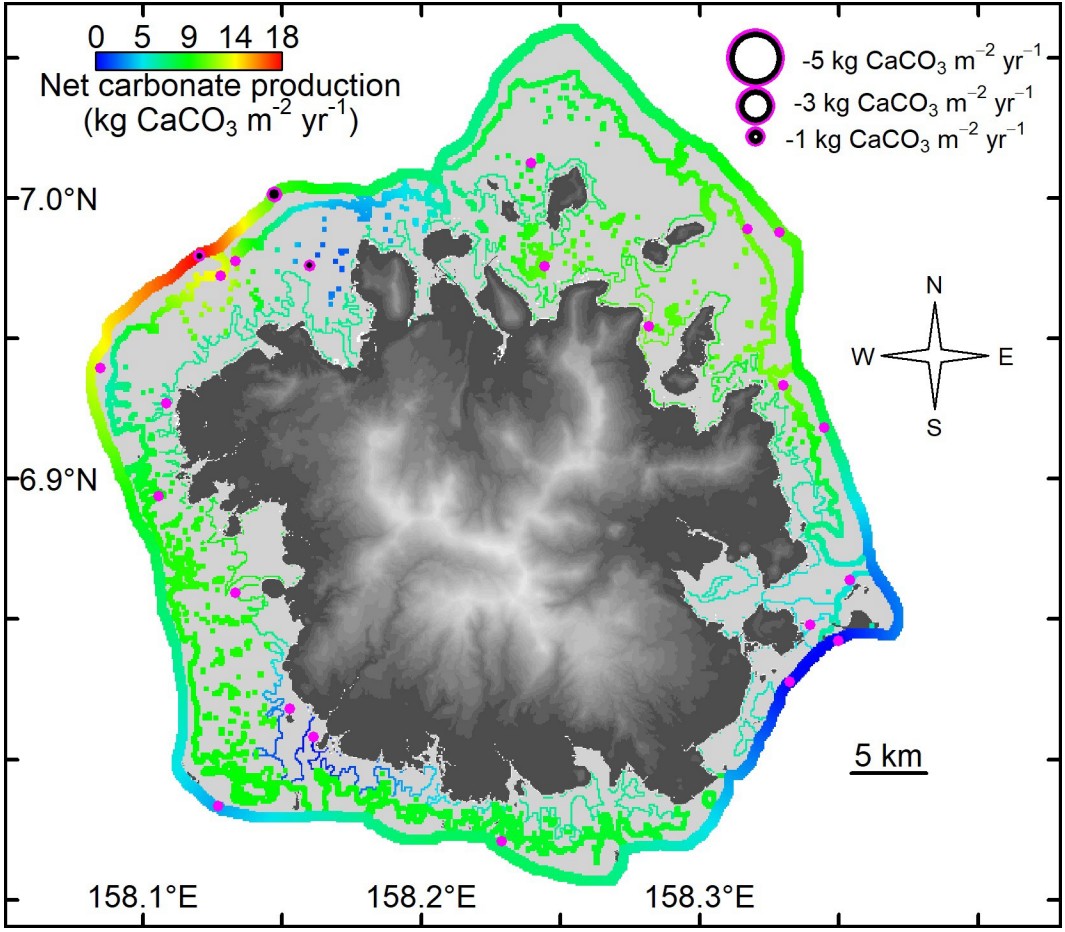

**Fig 6. The effect of *Acanthaster solaris* on the net shallow-water coral-reef carbonate production of Pohnpei, Federated States of Micronesia, 2018 where the size of the bubble is proportional to the carbonate reduction by *A. solaris*.** Plain magenta dots are sites that had no observed *A. solaris*. Base elevation map was plotted in R using raw 10 m Digital Elevation Model from https://pae-paha.pacioos.hawaii.edu/thredds/ncss/usgs_dem_10m_pohnpei/dataset.html.

rates of vertical accretion averaging 5.9 mm yr$^{-1}$, which is lower than most predicted rates of sea-level rise, even the conservative rates associated with RCP 4.5 by the year 2100 (Figure G in S1 File). Although there is some uncertainty in the live-coral-cover thresholds for the inner reefs of Pohnpei and Kosrae, these inner reefs on average require around 5.5% higher live coral cover than outer reefs to produce the same amount of carbonate (Fig 4). These results provide a strong conservation message that in order for nearshore reefs to have a chance to keep up with sea-level rise, it is critical to mitigate land-use discharge and pollution to near-shore shallow-water reefs. Without conserving these relatively sensitive, nearshore reefs our projections suggest that they would likely drown in the near future. Additionally, mitigating terrestrial runoff may also prevent large, persistent outbreaks of *Acanthaster* [21, 22].

Increases in the survival of *Acanthaster* brachiolaria-stage larvae have been associated with river discharge and elevated nutrient concentrations [21, 22]. Therefore, implementing management strategies on small Pacific islands that mitigate terrestrial discharge will not only reduce sedimentation stress on corals but will also effectively suppress chronically dense *Acanthaster* populations that reduce carbonate production. In Kosrae, *A. solaris* reduced carbonate production by on average 0.6 kg CaCO$_3$ m$^{-2}$ yr$^{-1}$, with a maximum reduction of 5.1 kg

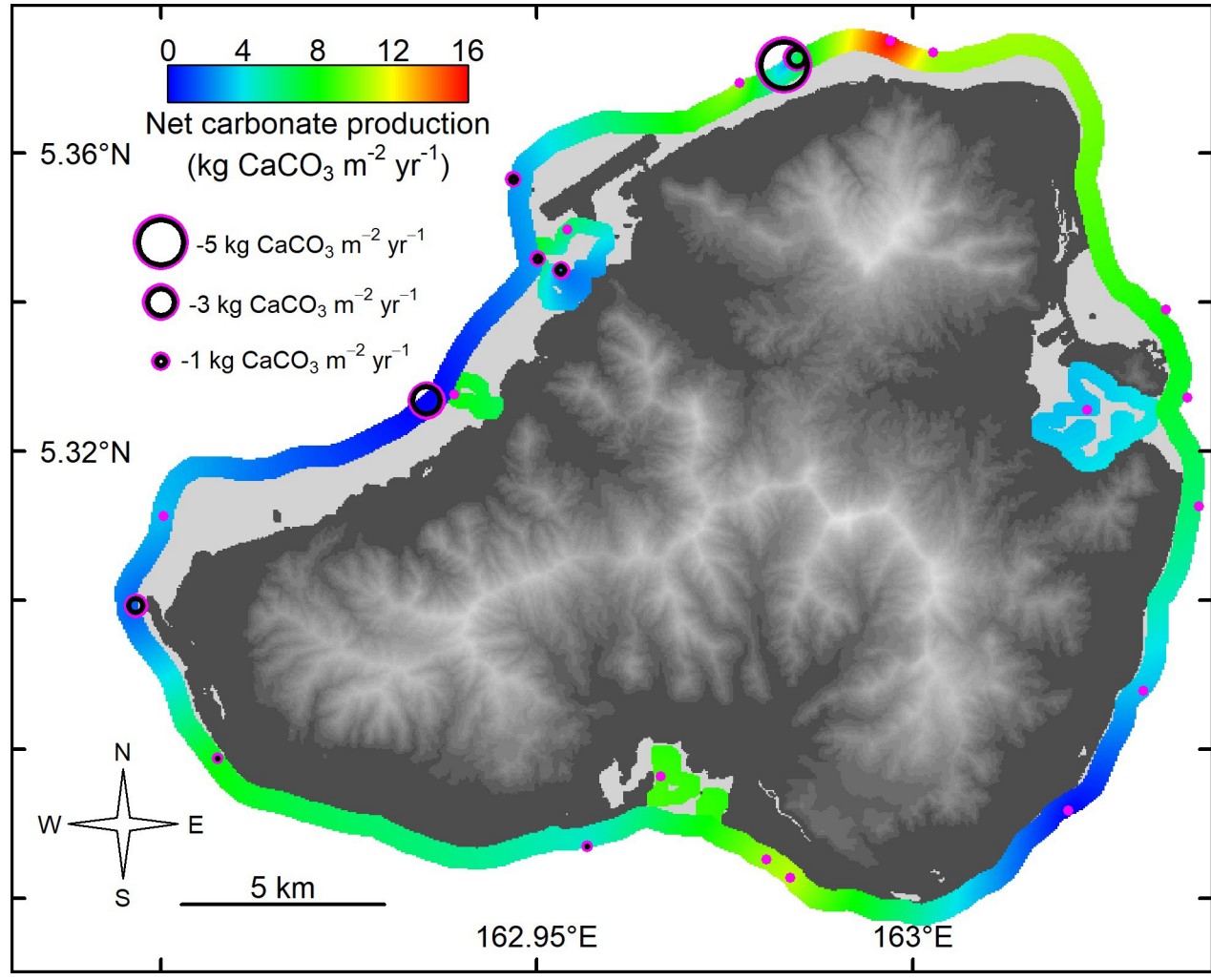

**Fig 7. The effect of *Acanthaster solaris* on the net shallow-water coral-reef carbonate production of Kosrae, Federated States of Micronesia, 2018 where the size of the bubble is proportional to the carbonate reduction by *A. solaris*.** Plain magenta dots are sites that had no observed *A. solaris*. Base elevation map was plotted in R using raw 10 m Digital Elevation Model from https://pae-paha.pacioos.hawaii.edu/thredds/ncss/usgs_dem_10m_kosrae/dataset.html.

$CaCO_3$ $m^{-2}$ $yr^{-1}$ when *A. solaris* were at high densities ($> 15$ individuals per 300 $m^2$). Although our estimates of the impact of *A. solaris* on carbonate production are novel, they still comprise a degree of uncertainty because of the assumptions underlying Eqs 14–17. Therefore, future improvements in these estimates can be made by examining these assumptions, which could include an adjustment for coral composition.

The coral species that were the most important contributors to carbonate production on both islands were *Porites rus* and *Porites lobata*, particularly on the shallow-water inner reefs. The most important contributors to carbonate production on the shallow-water outer reefs were encrusting *Montipora*, merulinids, and acroporids. There was a lack of patch reefs and large lagoons in Kosrae, therefore *Porites cylindrica* was less common on Kosrae than on Pohnpei. Yet, if the shallow-water outer reefs are unable to keep up with sea-level rise and are breached by offshore waves, the patch reefs of Pohnpei and the inner reefs of both islands will likely become more similar in coral composition to that of the outer reefs [47]. Still, whether

these reefs, even altered in composition, will be able to produce enough carbonate to keep up with sea-level rise is an open question.

The shallow-water coral-reef carbonate production rates measured on both Pohnpei and Kosrae are lower than the field estimates recorded farther west on Palau and Yap (~2.2 kg $CaCO_3$ m$^{-2}$ yr$^{-1}$ less, when averaged among habitat types) [17]. The lower carbonate production rates are most likely a result of reduced *Acropora* cover caused by the recent thermal-stress events on both Kosrae and Pohnpei in 2016 and 2017 (Peter Houk, pers. comm.). At the same time, similar thermal-stress events were not recorded in Palau and Yap. Thermal-stress events are known to significantly reduce a reefs' capacity to produce calcium carbonate [19], and under extreme events can temporarily reduce net accretion to negative values [18]. Although there are some studies on the net ecosystem calcification of coral reefs and the influence of coral bleaching on that process [47–51], more field studies are needed that examine (i) thermal-stress events and the dynamics of carbonate production through those events, and (ii) the rates of recovery of carbonate production from thermal-stress events.

The capacity of coral reefs to keep up with rising sea level is important for coastal residents and is particularly relevant to residents of low-lying islands who cannot move to higher elevations. Historically, healthy coral reefs have kept up with dynamic shifts in sea level through glacial-interglacial periods [52], yet disturbances to modern reefs are suppressing the capacity of coral reefs to produce enough carbonate [53] and protect island residents from storm-wave damage. In addition, drowned reefs will not be able to provide goods and services or support fisheries. If the coral species contributing to reef complexity and carbonate production are unable to persist under the stress of climate change then the coral reefs will not keep up with sea-level rise and drown.

## Supporting information

**S1 File. Seven supporting figures.** These figures include (A) a location map, (B) limit on the number of *Acanthaster solaris* sea stars, as a proportion of live coral cover (LCC) that a 100 m$^2$ shallow-water coral-reef habitat, (C and D) erosional kriged maps for both islands, and (E and F) erosional maps for parrotfishes for both islands, and (G) net vertical accretion split by strata compared to the rates of sea-level rise under representative concentration pathways 2.6, 4.5, 6.0, and 8.5 W m$^{-2}$.
(PDF)

**S1 Data. Data and R code.** Spreadsheet data for each site of the 48 sites on Kosrae and Pohnpei and all the R scripts that produced the manuscript figures.
(RAR)

## Acknowledgments

In Pohnpei, FSM, we would like to sincerely thank Alloise Malfitani and Valentina Permiakova, from the Pohnpei Dive Club, for their warm friendship and hospitality, together with Jerry Martins, Walden Lohn and Robinson Lowdaur for captaining our research vessel, and who all willingly shared their expert local knowledge. Thanks also go to Mark Figueras and staff from the Mangrove Bay Hotel for hosting us in Pohnpei. We would also like to thank Eugene Joseph, and his marine monitoring team for a collaborative opportunity at the Conservation Society of Pohnpei. In Kosrae, our research was made possible by our marvelous hosts Mark, Maria, and Ochi Stevens and staff from the Pacific Treelodge and Micronesian Ecodivers, together with Carlos J. Cianchini, Inston, and Salik Waguk of Kosrae Association of Tourism Operators, who all willingly shared their intimate knowledge of their island

environment and ensured our every need was met. We would also like to thank Andy George and his marine monitoring team at the Kosrae Conservation Society for a collaborative opportunity. Special thanks go to Sandra J van Woesik, Kelly J van Woesik, Liana J van Woesik, and Marina L Fleming for their invaluable voluntary field and lab assistance. We would also like to extend our thanks to Sandra J van Woesik and the anonymous reviewers for their editorial comments on the manuscript. This is contribution number 217 from the Institute for Global Ecology at the Florida Institute of Technology.

## Author Contributions

**Conceptualization:** Robert van Woesik.

**Data curation:** Robert van Woesik, Christopher William Cacciapaglia.

**Formal analysis:** Robert van Woesik, Christopher William Cacciapaglia.

**Funding acquisition:** Robert van Woesik.

**Investigation:** Robert van Woesik, Christopher William Cacciapaglia.

**Methodology:** Robert van Woesik, Christopher William Cacciapaglia.

**Project administration:** Robert van Woesik.

**Resources:** Robert van Woesik.

**Supervision:** Robert van Woesik.

**Validation:** Robert van Woesik, Christopher William Cacciapaglia.

**Visualization:** Robert van Woesik, Christopher William Cacciapaglia.

**Writing – original draft:** Robert van Woesik, Christopher William Cacciapaglia.

**Writing – review & editing:** Robert van Woesik, Christopher William Cacciapaglia.

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
