## [Decision Letter · Decision Letter 0]

26 Jun 2019

PONE-D-19-15515

Carbonate production of Micronesian reefs suppressed by thermal anomalies and Acanthaster as sea-level rises

PLOS ONE

Dear Dr van Woesik,

Thank you for submitting your manuscript to PLOS ONE. After careful consideration, we feel that it has merit but does not fully meet PLOS ONE’s publication criteria as it currently stands. Therefore, we invite you to submit a revised version of the manuscript that addresses the points raised during the review process.

As you will see, both reviewers provided constructive and valuable feedback. In both cases they were positive about the manuscript and felt that it was worthy of publication, however, they both raised concerns. In particular, there were concerns about the role of sediment influx in carbonate budgets, concerns about some of the data presentation and neglecting some literature on Acanthaster erosion.

We would appreciate receiving your revised manuscript within 30 days of receipt of this email. To enhance the reproducibility of your results, we recommend that if applicable you deposit your laboratory protocols in protocols.io, where a protocol can be assigned its own identifier (DOI) such that it can be cited independently in the future. For instructions see: http://journals.plos.org/plosone/s/submission-guidelines#loc-laboratory-protocols

We look forward to receiving your revised manuscript.

Kind regards,

James R. Guest, Ph.D.

Academic Editor

PLOS ONE

Journal Requirements:

4. We noticed you have some minor occurrence of overlapping text with the following previous publication(s), which needs to be addressed:

https://journals.plos.org/plosone/article?id=10.1371%2Fjournal.pone.0197077

In your revision ensure you cite all your sources (including your own works), and quote or rephrase any duplicated text outside the methods section. Further consideration is dependent on these concerns being addressed.

5.  We note that Figures 2, 3, 6 and 7 in your submission contain [map/satellite] images which may be copyrighted. All PLOS content is published under the Creative Commons Attribution License (CC BY 4.0), which means that the manuscript, images, and Supporting Information files will be freely available online, and any third party is permitted to access, download, copy, distribute, and use these materials in any way, even commercially, with proper attribution. For these reasons, we cannot publish previously copyrighted maps or satellite images created using proprietary data, such as Google software (Google Maps, Street View, and Earth). For more information, see our copyright guidelines: http://journals.plos.org/plosone/s/licenses-and-copyright.

1.    You may seek permission from the original copyright holder of Figures 2, 3, 6 and 7 to publish the content specifically under the CC BY 4.0 license.

Reviewers' comments:

Reviewer's Responses to Questions

**Comments to the Author**

1. Is the manuscript technically sound, and do the data support the conclusions?

Reviewer #1: Partly

Reviewer #2: Yes

2. Has the statistical analysis been performed appropriately and rigorously? 

Reviewer #1: Yes

Reviewer #2: Yes

3. Have the authors made all data underlying the findings in their manuscript fully available?

Reviewer #1: Yes

Reviewer #2: Yes

4. Is the manuscript presented in an intelligible fashion and written in standard English?

Reviewer #1: Yes

Reviewer #2: Yes

5. Review Comments to the Author

Reviewer #1: This is an overall well-written and well-executed study documenting the net carbonate production budgets for islands in Micronesia. Most significantly, the authors have pioneered spatial kriging within an excellent random, stratified plot design to determine net carbonate budgets around the perimeters of the entire islands. I believe this aspect of the study is an exceptional contribution to the field and applaud the authors for this contribution in using these methods to determine island scale carbonate production budgets. However, I have some concerns with how sedimentation and Acanthaster erosion were quantified and the language surrounding the capacity for reef accretion to keep up with sea level rise. I think this is nonetheless and important study and contribution to the field and hope the authors find the below comments helpful for improving the submitted manuscript prior to publication.

Sedimentation##################################################################

Lines 115–117/132–138: If sedimentation is high, how does that generate a negative term for the carbonate production equation? There’s more carbonate being dumped on the site so this should be positive… I see the clarification for this based on the influences of terrestrial sedimentation. However, if solid particles are settling on a reef then it’s adding material that will increase accretion. I agree that sedimentation could reduce calcification rates by some factor and organics could increase CaCO3 dissolution rates; however, I disagree with utilizing a baseline –0.4 kg CaCO3/m2/y as an appropriate adjustment for such affects considering that there is a material flux onto the reef. I think the sedimentation section should more accurately reflect what was actually measured in this study and the authors are welcome speculate that sedimentation decreased calcification rates nearshore if they wish to include such a conclusion in the manuscript but I do not think they should directly quantify the –0.4 kg CaCO3/m2/y in their budgets owing to sedimentation being present.

Acanthaster####################################################################

Lines 189-217: I appreciate the author's efforts to include Acanthaster erorion in this study however this section neglects quite a bit of the literature on the effects of Acanthaster on coral reef growth and should therefore be restructured. For example, I see no evidence in the present manuscript that Acanthaster actually consumes the coral calcium carbonate. The authors should note that Eakin (1996) found that Acanthaster digests tissue without destroying skeletons and Glynn (1973) previously assessed densities of Acanthaster that would overwhelm the ability for corals to persist. Lastly, Henderson and Walbran (1992) found that Acanthaster CaCO3 skeletal fragments were preserved in reef sediments and therefore added to reef CaCO3 production. I recognize that there is an influence of reducing coral cover owing to consumption of corals by Acanthaster that would drive decreases in CaCO3 production indirectly through the reduction in corals sensu Glynn (1973); however, the equations presented in this section make it seem as though Acanthaster is actually consuming CaCO3 in an erosional sense- a finding that the aforementioned papers would contradict and the present study does not provide sufficient quantifiable evidence otherwise. If the goal is to estimate the loss in CaCO3 production owing to Acanthaster- why not estimate a mean loss rate of corals per annual as per your equations and apply this loss in coral cover to the coral carbonate production equation to estimate the lost capacity for producing calcium carbonate by corals? However, if this is the study design then there should be some consideration of coral recruitment and whether or not the coral cover is in a quasi-steady state through time.

Glynn, P. W. (1973). Acanthaster: effect on coral reef growth in Panama. Science, 180(4085), 504-506.

Eakin, C. M. (1996). Where have all the carbonates gone? A model comparison of calcium carbonate budgets before and after the 1982–1983 El Nino at Uva Island in the eastern Pacific. Coral Reefs, 15(2), 109-119.

Henderson, R. A., & Walbran, P. D. (1992). Interpretation of the fossil record of Acanthaster planci from the Great Barrier Reef: a reply to criticism. Coral Reefs, 11(2), 95-101.

Accretion######################################################################

Line 58 (and throughout): Accretion generally refers to vertical accretion (mm/y) whereas net carbonate production generally refers to (kg CaCO3/m^2/y). Because Perry et al (2013) calculates both accretion and net carbonate production rates, I think “accretion” should be switched to “net carbonate production” to be more consistent with the numbers from Perry et al (2013). It would be good to be consistent throughout the manuscript with respect to the use of accretion vs. net carbonate production terms.

Line 219: What is the value for alpha in this study? I visited the previous van Woesik and Cacciapaglia (2018) publication and the alpha term derived therein referred me to the supplementary data of that manuscript where it’s a single value of -0.01949. Is this the value used in the present study? If so please just put this value in the text in place of requiring a reader to jump through so many hoops to determine a single number. Additionally, how does this calculation compare to the previously used values derived by Kinsey (1985) assuming a density of 2.9 g cm^-3 for calcium carbonate and average porosity of 50% and leveraged by Perry et al (2018) where they used 30-70% porosity with 50% reincorporation of parrotfish derived sediment back into the reef framework?

Kinsey (1985) Metabolism, calcification and carbon production. Proceedings of the fifth international coral reef congress, Tahiti, Vol 4.

Perry, C. T., Alvarez-Filip, L., Graham, N. A., Mumby, P. J., Wilson, S. K., Kench, P. S., ... & Januchowski-Hartley, F. (2018). Loss of coral reef growth capacity to track future increases in sea level. Nature, 558(7710), 396.

Lines 331-345: Perhaps I am confused about how the authors dealt with sedimentation in this study, but if the net CaCO3 sediment transport to the inner reefs (and microbioerosion/CaCO3 dissolution) was not considered than in my opinion these statements are likely too strongly stated. I caution making strong statements on such matters when the accretion rates of the inner reefs weren’t directly quantified in this study (e.g., as in Yates et al 2017) and the budgets presented here do not appear to fully cover all of the terms necessary to quantify net carbonate accumulation (i.e., see Kleypas et al [2001]). Since accretion is estimated from the budgets and the relationship between net carbonate production budgets and accretion rate was based on just a few points, there is quite a bit of uncertainty in this value that should temper the conclusions drawn from it especially considering that such statements potentially impact human population centers associated with these reefs in Micronesia and set the precedent for the same standard elsewhere.

Perry, C. T., Kench, P. S., Smithers, S. G., Riegl, B. R., Gulliver, P., & Daniells, J. J. (2017). Terrigenous sediment-dominated reef platform infilling: an unexpected precursor to reef island formation and a test of the reef platform size–island age model in the Pacific. Coral Reefs, 36(3), 1013-1021.

Yates, K. K., Zawada, D. G., Smiley, N. A., & Tiling-Range, G. (2017). Divergence of seafloor elevation and sea level rise in coral reef ecosystems. Biogeosciences, 14(6), 1739.

Perry, C. T., Morgan, K. M., & Yarlett, R. T. (2017). Reef habitat type and spatial extent as interacting controls on platform-scale carbonate budgets. Frontiers in Marine Science, 4, 185.

Kleypas, J. A., Buddemeier, R. W., & Gattuso, J. P. (2001). The future of coral reefs in an age of global change. International Journal of Earth Sciences, 90(2), 426-437.

Eyre, B. D., Andersson, A. J., & Cyronak, T. (2014). Benthic coral reef calcium carbonate dissolution in an acidifying ocean. Nature Climate Change, 4(11), 969.

Coral bleaching and reef-scale calcification##########################################

Line 375: There is also a growing literature on coral reef net ecosystem calcification and coral bleaching that provide critical information to the dynamics of carbonate production and thermal stress events and could be cited here. Please see the following papers:

Kayanne, H., Hata, H., Kudo, S., Yamano, H., Watanabe, A., Ikeda, Y., ... & Saito, H. (2005). Seasonal and bleaching‐induced changes in coral reef metabolism and CO2 flux. Global Biogeochemical Cycles, 19(3).

Watanabe, A., Kayanne, H., Hata, H., Kudo, S., Nozaki, K., Kato, K., ... & Yamano, H. (2006). Analysis of the seawater CO2 system in the barrier reef‐lagoon system of Palau using total alkalinity‐dissolved inorganic carbon diagrams. Limnology and Oceanography, 51(4), 1614-1628.

DeCarlo, T. M., Cohen, A. L., Wong, G. T., Shiah, F. K., Lentz, S. J., Davis, K. A., ... & Lohmann, P. (2017). Community production modulates coral reef pH and the sensitivity of ecosystem calcification to ocean acidification. Journal of Geophysical Research: Oceans, 122(1), 745-761.

Courtney, T. A., De Carlo, E. H., Page, H. N., Bahr, K. D., Barro, A., Howins, N., ... & Andersson, A. J. (2018). Recovery of reef‐scale calcification following a bleaching event in Kāne'ohe Bay, Hawai'i. Limnology and Oceanography Letters, 3(1), 1-9.

McMahon, A., Santos, I. R., Schulz, K. G., Scott, A., Silverman, J., Davis, K. L., & Maher, D. T. (2019). Coral reef calcification and production after the 2016 bleaching event at Lizard Island, Great Barrier Reef. Journal of Geophysical Research: Oceans.

Minor comments################################################################

Lines 88–95: excellent site selection design!!!

Lines 129-130: Isn’t the gross carbonate production in g/cm^2/y and the resulting units after conversion kg/m^2/y?

Please maintain consistency with units as “y” and “yr” are used interchangeably throughout the manuscript

The erosion section and equations therein are quite similar to Perry et al. (2012) and should be cited as such where appropriate.

Line 199: Shouldn’t the kg/m^2 be per year?

Line 256: Couldn’t this just in part be driven by the somewhat arbitrarily assigned -0.4 kg/m2/y rate applied to the high sedimentation areas? Alternatively, how did the authors account for the very large expanses of sand that typically characterize inshore lagoonal environments?

Line 311: add space to “reefs,and”

Line 362-364: This is an excellent point and is a feature of geological perspectives of coral reefs. For example, please see Engels (2004) and Chapter 6 Montaggioni and Braithwaite (2009) for potential citations for this.

Engels, M. S., Fletcher III, C. H., Field, M. E., Storlazzi, C. D., Grossman, E. E., Rooney, J. J., ... & Glenn, C. (2004). Holocene reef accretion: southwest Molokai, Hawaii, USA. Journal of Sedimentary Research, 74(2), 255-269.

Montaggioni, L. F., & Braithwaite, C. J. (2009). Quaternary coral reef systems: history, development processes and controlling factors (Vol. 5). Elsevier.

Reviewer #2: This manuscript provides carbonate budgets for two coral reefs in Micronesia, a region that is likely to be highly impacted by future sea-level rise. The manuscript is well-written, the methods are sound, and the results provide much needed data from an understudied region. Most of my comments are relatively minor, but I do have a couple of more substantial points, that I would like to see the authors address before publication. Both are outlined in the document attached.

6. PLOS authors have the option to publish the peer review history of their article (what does this mean?). If published, this will include your full peer review and any attached files.

Reviewer #1: No

Reviewer #2: No

---

## [Editor Report · Decision Letter 1]

24 Oct 2019

Carbonate production of Micronesian reefs suppressed by thermal anomalies and Acanthaster as sea-level rises

PONE-D-19-15515R1

Dear Dr. van Woesik,

We are pleased to inform you that your manuscript has been judged scientifically suitable for publication and will be formally accepted for publication once it complies with all outstanding technical requirements.

With kind regards,

James R. Guest, Ph.D.

Academic Editor

PLOS ONE

Additional Editor Comments (optional):

I apologize for the delay in getting a final response to you on your revised manuscript. I have been through your responses to the reviewers initial comments and am completely satisfied that you have addressed all of them satisfactorily. I have no hesitation in recommending publication of the revised draft.

Reviewers' comments:

No additional comments

---

## [Editor Report · Acceptance letter]

5 Nov 2019

PONE-D-19-15515R1 

Carbonate production of Micronesian reefs suppressed by thermal anomalies and *Acanthaster* as sea-level rises 

Dear Dr. van Woesik:

I am pleased to inform you that your manuscript has been deemed suitable for publication in PLOS ONE. Congratulations! Your manuscript is now with our production department. 

With kind regards,

on behalf of

Dr. James R. Guest 

Academic Editor

PLOS ONE